# Analysis of Construction Networks and Structural Characteristics of Pearl River Delta and Surrounding Cities Based on Multiple Connections

**Shengdong Nie and Hengkai Li \***

School of Civil and Surveying and Mapping Engineering, Jiangxi University of Science and Technology, Ganzhou 341000, China; 6120220148@mail.jxust.edu.cn
* Correspondence: giskai@126.com

**Abstract:** The Pearl River Delta (PRD) is one of three world-class city clusters in China, which is important for the strategical deployment of the national "Belt and Road". Based on nighttime lighting data, Baidu index, and train stopping times, this study analyzed the network of spatial patterns and structural evolution of the PRD and surrounding cities via social network analysis and dynamic network visualization, providing new perspectives and ideas for the study of intercity linkages and urban networks. The results provide decision support to the government for urban cluster planning. From 2014 to 2020, the economic network evolved from a uniaxial structure to an "inverted V" structure. The transportation network evolved from a uniaxial structure to a "△" structure. The information network did not show any obvious structural changes during its development, except for a star-shaped radial structure. The PRD city cluster and its surrounding cities exhibited a spatially non-uniform distribution in terms of spatial connections. The total connections between Guangzhou and Foshan and the surrounding cities in terms of economic, transportation, and information functions account for 30%, 28%, and 10% of the total urban connections, respectively. The graph entropy growth rates of the PRD city cluster and surrounding cities in economic, transportation, and information networks from 2014 to 2020 were 39.9%, 115.4%, and 5.1%, respectively. The network structures of economic and transportation networks stabilized eventually. The information network structures are stable.

**Keywords:** Pearl River Delta; urban agglomeration; information network; transportation network; economic network; spatial analysis; geoinformation

## 1. Introduction

The study of urban networks originated with the concept of "mobile space" [1]. Castells proposed the concept of "mobile space" and proposed that urban development does not depend on local static functions, which provided a new perspective on the study of urban areas [2]. The city cannot be independent of the regional urban system, and its development cannot be separated from the interrelated urban system [3]. After the 1990s, the acceleration of globalization and informatization led to the rapid development of urban systems around the world. The promotion of high-speed infrastructure and information networks led to the formation of spatiotemporal sharing networks between cities through intercity connections at multiple spatial scales and levels [4]. The increasingly close communication between cities contributed to the evolution of regional association patterns based on the mobility and interdependence of elements from a central place shaped by "space of place" to a polycentric network model nurtured by "space of mobility" [5,6].

Urban agglomerations are an important spatial form underlying the shift in the global economic center of gravity and are attributed to advanced urbanization and industrialization in China [7]. Since reform and opening up, China has experienced the largest and

fastest urbanization in the history of the world and has gradually formed a new developmental model of urbanization with urban agglomerations as the main form [8]. Currently, with the rapid development of urban networking and the related strategic vision of the country, the important spatial form of regional development, an urban agglomeration, and the connection between cities in urban agglomerations have become a research hot spot [9].

Capello [10] defines the "city network paradigm" as a network in which cities achieve economies of scale through complementary interactions, collaborative actions, and cooperative efforts. Network theory enables the examination of horizontal relationships between cities, thus providing an appropriate theoretical framework to overcome the limitations of the central location model. The introduction of the "city network paradigm" offers a theoretical framework for studying the spatial structure of megacities [11].

In previous studies on urban network structures, researchers have utilized the concept of the urban network paradigm to investigate various urban regions, such as megacities [12], megaregions [13], megacity regions [14], global cities [15], and urban agglomeration [16]. The spatial structure of urban networks, as a spatial distribution pattern resulting from the interactions among urban nodes [17], emphasizes the horizontal and complementary integration between urban regions based on the establishment of transportation networks and intercity complementarity [10,18]. The urban network paradigm is aligned with the functional perspective of a multi-centered spatial structure, as it highlights the functional connections and interactions among multiple central regions within urban regions [19].

Many studies have employed the urban network paradigm to investigate various aspects of spatial structure in megacities. For instance, Marull, Font, and Boix [20] utilized DMSP-OLS nighttime lights and urban connections as nodes, while major roads and railway facilities were considered edges. They measured the spatial structure of European megaregions using four indicators: complexity, multicentricity, efficiency, and stability. Taubenbock and Wiesner [21] focused on assessing spatial connectivity between cities based on the continuity of settlement patterns and differentiated types of global megacities. Additionally, some studies have analyzed the community structure of integrated regions through the lens of urban network theory. For example, Baek and Joo [19] analyzed the regional centrality of the Bu-Ul-Gyeong (BUG) megacity at various spatial scales using mobile flow data from 2019 to 2020.

Furthermore, numerous scholars have constructed urban networks from a functional perspective using various types of data to analyze urban spatial structures. These include traffic flow data, enterprise organizational associations, and information flow data. In the field of traffic flow, relevant studies have primarily focused on air passenger flows [22], high-speed rail networks [23,24], and intercity railway networks [25]. In the field of enterprise organizational flows, recent research has extensively conducted empirical analyses of global/China/regional/city networks by expanding the types of enterprises and optimizing network models [26,27]. In the domain of information flows, the advancement of information networks has led to the emergence of various new types of data, including hot search index data, social location data, migration data, and point of interest (POI) data. Many scholars have utilized these data to analyze and study the structures of information networks at different scales [6,28–30].

Based on the aforementioned studies, additional data are available for selection in urban networks. However, the data are limited by different perspectives and types based on category, detail, and accuracy, and the existing research objects are mostly related to a single type of network within a region [31]. Urban network challenges are systemic, complex, and multifaceted, and no comprehensive conclusions can be drawn based on only a single type of network [32]. Cities cooperate and connect through economic, informational, traffic, and technological flows at various spatial scales and per specific requirements, ultimately forming an indivisible and organic whole [33]. Considering the availability of data, we have integrated the three aspects of economy, information, and transport, intending to characterize the connections between cities more comprehensively. In addition,

existing studies of urban networks are mostly undirected weighted networks, ignoring the asymmetric flow of factors [34].

The Pearl River Delta (PRD) region has served as a "testing ground" for China's reform and opening up, playing a pivotal role in the economic development of Guangdong Province over the past three decades. It has emerged as a formidable engine driving China's economic and social progress. The Guangdong-Hong Kong-Macao Greater Bay Area Development Plan suggests that the Pan-PRD (PPRD) region should be used as the vast developmental hinterland. The Guangdong-Hong Kong-Macao Greater Bay Area should be built into a dynamic world-class city cluster, an endeavor of significant importance in the construction of the "One Belt, One Road" initiative. Considering the geographical proximity, effective utilization of urban resources in the PPRD region should be based on strengthening the connection and interaction between the PRD city cluster and the cities in the PPRD region. However, few studies explore the enhancement of exchanges and interactions between the PRD city cluster and the cities in the PPRD region. Therefore, it is crucial to investigate the intercity linkages between the PRD city cluster and the neighboring inland cities. Existing studies primarily concentrate on aspects such as the economy [35], industry [36], and port attractiveness [37]. These studies rarely provide an in-depth analysis of the linkages between PRD urban agglomerations and the surrounding cities from multiple perspectives. In addition, they lack differentiation between intercity linkages. In October 2014, the governments of nine provinces and two regions within the Pan-Pearl River Delta (PPRD) region collaboratively formulated the Joint Declaration on Deepening Cooperation in the PPRD Region (2015–2025), aiming to facilitate deeper cooperation on a wide range of issues. In light of this initiative, we selected the time period from 2014 to 2020 for our study, intending to examine the evolution of the urban function network structure between cities in the PRD and surrounding inland areas following the declaration's formulation.

In previous studies on urban agglomerations, the intensity of economic linkage has been assessed using traditional statistics such as GDP and population [29]. Similarly, the intensity of transportation linkage has been evaluated based on train frequency data [38], and information linkage intensity is measured using the Baidu index [39].

Accordingly, this study used traditional statistical data, NPP/VIIRS Nighttime Light (NTL) data, urban road network data, Baidu index, and train stopping times to construct a directional weighted network to explore the development and evolution of intercity links and network spatial structure between the PRD city cluster and the hinterland cities from economic, information, and transportation perspectives. The research framework is illustrated in Figure 1. The findings provide a deeper insight into the functional links and spatial structure involving the PRD city cluster and the inland hinterland cities. The study facilitates regional planning and construction of the PRD city cluster.

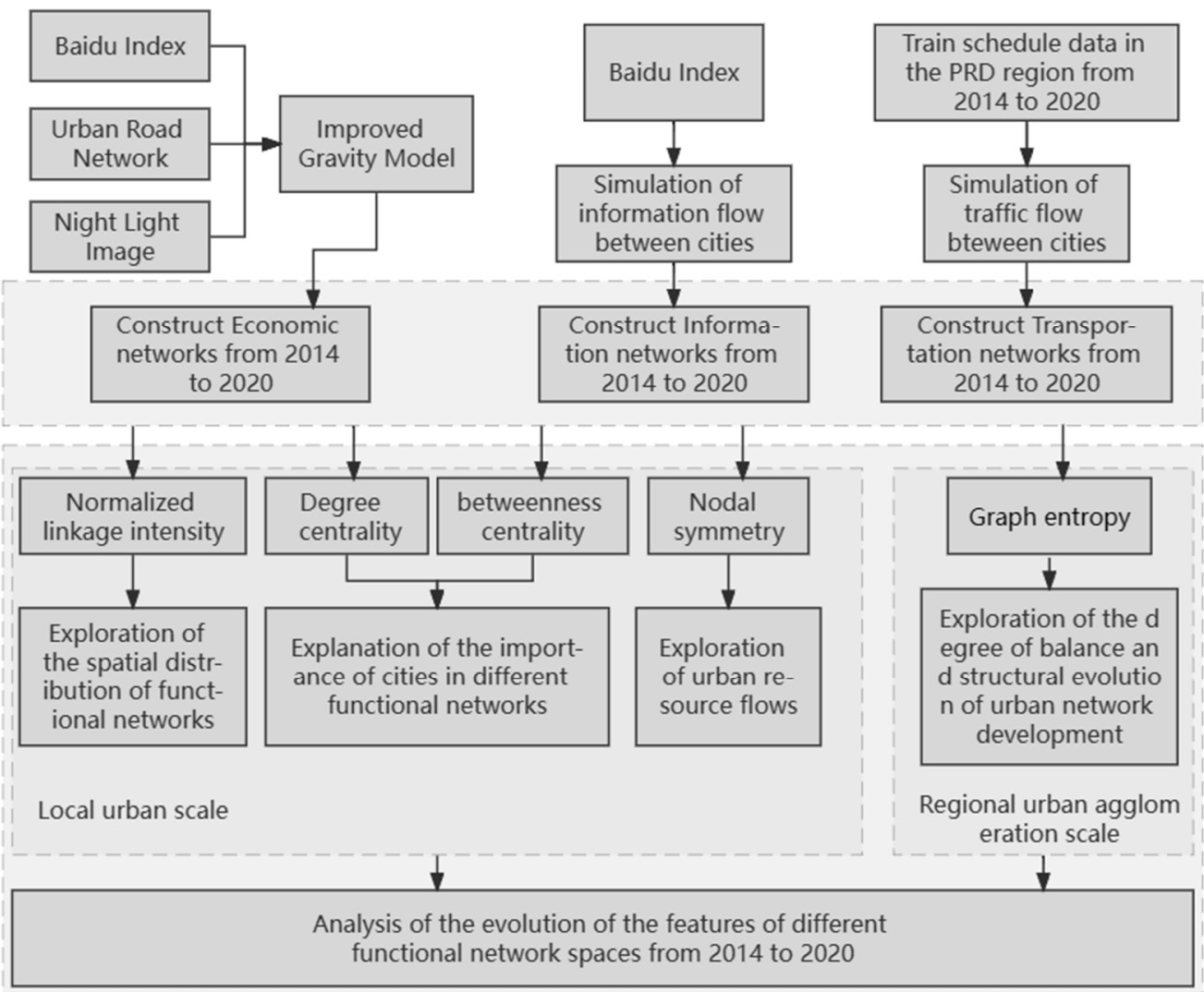

**Figure 1.** Research framework for this study.

## 2. Study Area

China's 13th Five-Year Plan proposes to build 19 city clusters nationwide, including the PRD, one of the three world-class city clusters in China [40]. The Pearl River Delta city cluster encompasses nine cities: Guangzhou, Foshan, Shenzhen, Dongguan, Huizhou, Zhaoqing, Jiangmen, Zhongshan, and Zhuhai. The cluster covers a combined land area of 55,000 square kilometers. As of 2020, the resident population of the Pearl River Delta city cluster had reached 78,235,400. In addition, the cluster's economy accounts for 8.86% of the country's total economy. It is one of the most economically dynamic regions in the Asia-Pacific region and the gateway to the south of China for the outside world, with development expanding to South, Central, and Southwest China. To more effectively study the exchanges and interactions between PRD and Pan-PRD cities, we extend the scope of the study to the entire Guangdong Province and the prefecture-level cities adjacent to the administrative boundary of Guangdong Province. The specific regional scope of the study is shown in Figure 2.

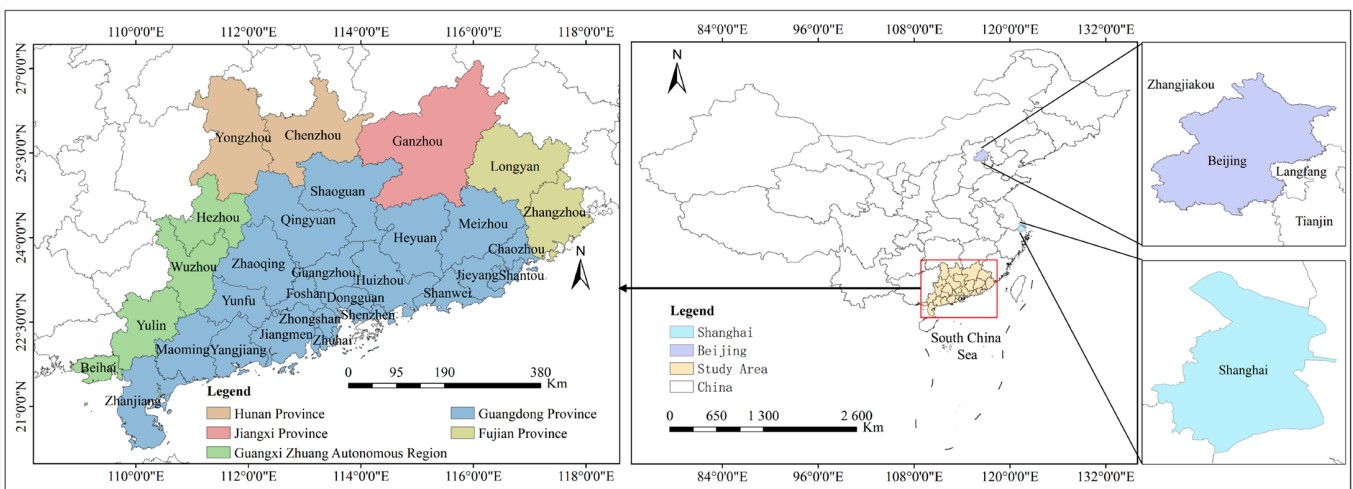

**Figure 2.** Administrative boundaries of Guangdong Province and its neighboring prefecture-level cities.

## 3. Materials and Methods

### 3.1. Data Source

The research data include statistical, traffic vector, and NTL data, as well as Baidu index and train stopping time data, over three years: 2014, 2017, and 2020. The statistical data were obtained from publicly available statistical yearbooks, primarily the corresponding year's China City Statistical Yearbook. Traffic vector data include highways, expressways, and national, provincial, and county roads in the study area of the corresponding year, as well as data from the official website of OpenStreetMap. NTL data (NNP/VIIRS Annual VNL V2 500 m × 500 m) were derived from the Group on Earth Observations. The NTL data for 30 study units were extracted, and the maximum image brightness of three well-developed cities (Guangzhou, Shanghai, and Beijing) was used as the annual benchmark. Based on Guan's study [41], we chose a benchmark value of $0.3 \times 10^{-9}$ W·cm$^{-2}$·Sr$^{-1}$ for the abnormally low value of the night light data. Outliers were removed. The train stopping time data were used to measure the intensity of traffic flow between cities based on the established train schedules and the Luton schedules. Since train trips do not change within a short period of time, we selected train stopping times on any day of the study year as the study data. However, the data for Yangjiang were unavailable in 2014 and 2017, while the 2014 data were missing for Yunfu. Therefore, this study used the 2015 train stop data from train stations in Yunfu and the 2018 data from Yangjiang stations as substitutes. The Baidu index partially represents the intensity of information flow between cities. The data were collected from 1 January to 31 December of the respective research year. Detailed information is shown in Table 1:

**Table 1.** Data sources used in the study.

| Data | Year | Spatial Resolution | Data Source |
|---|---|---|---|
| Road Network Data | 2014, 2017, 2020 | -- | https://www.openhistoricalmap.org (accessed on 3 August 2022). |
| Train time data | 2014, 2017, 2020 | -- | http://www.smskb.com (accessed on 10 December 2013). http://www.lltskb.com/ (accessed on 28 December 2017 and 28 June 2020). |
| Night lighting data | 2014, 2017, 2020 | 0.5 km | https://eogdata.mines.edu/products/vnl (accessed on 3 August 2022). |
| Baidu Index | 2014, 2017, 2020 | -- | https://index.baidu.com (accessed on 3 August 2022). |
| GDP | 2014, 2017, 2020 | -- | http://www.stats.gov.cn/sj/ndsj/ (accessed on 3 August 2022). |
| Vector Boundary | 2017 | -- | https://www.webmap.cn/main.do?method=index (accessed on 3 August 2022). |

*3.2. Research Methods*

3.2.1. City Structure Network Construction Method

City Economic Network

The weights of the nodal edges in the economic network are expressed by the strength of the economic ties between the corresponding cities. Zipf proposed the economic gravity model to describe the relationship between cities [42]. It is widely used in the fields of regional trade, resource flow, and urban space research. The economic gravity model can be mathematically expressed as follows:

$$E_{ij} = g \frac{M_i M_j}{D_{ij}^2} \tag{1}$$

where $E_{ij}$ represents the degree of economic linkage between city $i$ and city $j$; $M_i$ and $M_j$ represent the quality indicators of city $i$ and city $j$, respectively; $D_{ij}$ denotes the distance between city $i$ and city $j$, and $g$ is the gravitational coefficient.

The traditional economic gravity model relies on conventional statistical data and Euclidean distance between cities as parameters. Considering the complexity of urban systems, statistical data and Euclidean distance alone cannot truly and accurately reflect the economic linkages between cities. Therefore, this study improved the city quality, city distance, and gravitational coefficients of the traditional economic gravity model to portray a more realistic strength of economic linkages between cities.

First, the improvement of urban quality parameters was studied. Previous studies have commonly relied on conventional statistics, such as national and regional gross domestic product (GDP) and population, as indicators of the economic importance of cities. Traditional statistical data has limitations in providing a comprehensive and detailed representation of intra-regional economic disparities. On the other hand, nighttime light data is not bound by geographical boundaries and offers several advantages, such as high objectivity, strong timeliness, extensive temporal coverage, and broad spatial coverage. Utilizing Nighttime Light (NTL) data to estimate local economic activity proves to be a cost-effective, objective, and highly accurate approach [43]. NTL data exhibits a strong correlation with local economic activities [44]. We reviewed the studies of Chen Jin, Pan Jinghu, Zhuo Li, and others on light indices [45–47] and used the urban NTL index to represent the urban quality index of the improved gravity model. Currently, the commonly employed NTL indices include the Total Night Light Index (TNLI), Average Night Light Index (ANLI), Average Night Light Intensity (I), Night Lighting Area Ratio (S), Linear Weighted Night Light Composite Index (LNLI), and Comprehensive Night Light Index (CNLI). The mathematical expressions of these indicators are provided in Table 2. We first validated the linear correlation between GDP and the lighting data indicators by considering the GDP of 30 cities. Based on this validation, we calculated the Pearson linear correlation coefficients for six lighting data indicators: TNLI, ANLI, I, S, LNLI, and CNLI. LNLI is derived from I and S, with weight values borrowed from Chen Jin's study on light indices to determine urbanization levels [44], set at 0.8 and 0.2, respectively. The formulas, Pearson correlation coefficient, and Statistical significance level (*p*-value) results are presented in Table 2. Based on the research findings, the optimal city quality indicator was determined.

In the equation presented in Table 2, $DN_i$ and $n_i$ denote the value and number of grayscale pixels at level $i$ in the study area, respectively; $N$ and $A_N$ represent the total number of pixels and the occupied area in the study area within the interval of $(0, DN_M]$, respectively; and $A$ denotes the area occupied by the study area.

Accordingly, in the case of a strong correlation between TNLI and GDP, the TNLI index represents the quality of cities and improves the gravity model to determine the strength of economic ties.

**Table 2.** The formula for calculating the lighting index and its correlation coefficient with GDP.

| Night Lighting Index | Calculation Formulas | 2014 | | 2017 | | 2020 | |
|---|---|---|---|---|---|---|---|
| | | Correlation | $p$ | Correlation | $p$ | Correlation | $p$ |
| Linear weighted Night Light Composite Index (LNLI) | $LNLI = 0.8 \times I + 0.2 \times S$ | 0.653 | $9.20 \times 10^{-5}$ | 0.591 | $5.85 \times 10^{-4}$ | 0.579 | $8.03 \times 10^{-4}$ |
| Average Night Light Intensity (I) | $I = \sum\limits_{i=1}^{DN_M} \frac{DN_i \cdot n_i}{DN_M \cdot N}$ | 0.630 | $1.92 \times 10^{-4}$ | 0.532 | $2.47 \times 10^{-3}$ | 0.542 | $1.99 \times 10^{-3}$ |
| Night Lighting Area Ratio (S) | $S = \frac{A_N}{A}$ | 0.608 | $3.70 \times 10^{-4}$ | 0.577 | $8.43 \times 10^{-4}$ | 0.552 | $1.58 \times 10^{-3}$ |
| Comprehensive Night Light Index (CNLI) | $CNLI = I \cdot S$ | 0.653 | $9.20 \times 10^{-5}$ | 0.592 | $5.70 \times 10^{-4}$ | 0.583 | $7.23 \times 10^{-4}$ |
| Total Night Light Index (TNLI) | $TNLI = \sum\limits_{i=1}^{n} DN_i$ | 0.807 | $6.96 \times 10^{-8}$ | 0.736 | $4.00 \times 10^{-6}$ | 0.647 | $1.13 \times 10^{-4}$ |
| Average Night Light Index (ANLI) | $ANLI = \sum\limits_{i=1}^{n} DN_i / n$ | 0.702 | $1.60 \times 10^{-5}$ | 0.675 | $4.20 \times 10^{-5}$ | 0.673 | $4.50 \times 10^{-5}$ |

Second, the improved urban distance was analyzed. Previous studies mostly used Euclidean distance to characterize the distance between cities. However, the morphology and density of urban road networks undergo significant changes over time. A city's road network can effectively represent the traffic conditions within the city, and the density of the road network can, to some extent, reflect the convenience of urban transportation. The Pearl River Delta urban agglomeration and surrounding cities experienced substantial changes in the morphology of their road networks during the study period. Using only the Euclidean distance between cities to represent their distance cannot accurately depict the developmental stages of cities.

For a given land transport distance, the higher the transportation convenience, the lower the transportation cost. Convenient transportation not only reduces transportation losses but also shortens the transportation process and reduces transportation risks, thereby increasing the potential for trade profitability from multiple aspects. Luo's study [48] constructed a "Transportation Convenience Index" to represent the degree of transportation convenience related to transportation costs. This index measures the convenience of transportation based on the density of road and railway transportation networks. Building upon previous research, this study uses the "Transportation Convenience Index", constructed based on road network density, to calculate the economic distance between two cities. The improved gravity model replaces the Euclidean distance with the economic distance between cities. Road network density is represented by the ratio of the total length of regional road networks to the regional area. The formula for calculating economic distance [8] is as follows:

$$D_{ij} = L_{ij}(1 + e^{-\sqrt{\rho_i \rho_j}}) \qquad (2)$$

where $\rho$ represents the road network density of the city; $e$ stands for Euler's constant; $L_{ij}$ and $D_{ij}$ represents the Euclidean distance and economic distance between city $i$ and $j$ city, respectively.

Third, the gravitational coefficients of the gravity model were improved. The majority of advancements in gravity models primarily focus on city quality and city distance. Limited research has addressed the enhancement of gravitational coefficients in gravity models. Furthermore, in the rare instances where such improvements were made, they were primarily based on an economic perspective utilizing traditional statistics. Baidu is the world's largest Chinese search engine and the most used search engine by Chinese Internet users. Employing the Baidu index for specific keyword searches enables a partial reflection of Internet users' psychological needs and plays a crucial role in behavior prediction [49]. The study utilized the Baidu index, which captures the search behavior of the majority of Internet users and employed search volume as the primary statistical indicator to construct the population mobility trend index. In China, we currently divide population mobility into two categories: short-term and long-term. Income levels and housing costs are long-term

mobility factors, while travel and business trips are short-term mobility factors. Drawing on other studies [34], this study selected "Recruitment + Jobs" as the keyword to capture urban income-related traffic, "Price + Lease" as the keyword to capture urban cost-of-living-related traffic, and finally "Map + Weather" as the supplementary keyword to capture trends in population movement based on short-term population mobility. The data variable population movement trend was constructed and $X_{ij}$ was defined as the trend of population movement from city $i$ to city $j$, which can be expressed mathematically as:

$$X_{ij} = \sqrt[3]{Hou_{ij} \times Job_{ij} \times MAW_{ij}}(i \neq j) \tag{3}$$

where $Hou_{ij}$ represents the search index of city-based Internet users for the city keyword " Price + Lease"; $Job_{ij}$ represents the search index of city-based Internet users for the city keyword "Recruitment + Jobs"; and $MAW_{ij}$ represents the search index of city-based Internet users for the city keyword " Map + Weather".

In summary, this study utilized the total NTL index of cities to assess the quality of cities, the population flows to capture the direction of urban gravity, and the economic distance between cities to measure urban distance. Thus, a gravity model was developed based on these factors, enhancing the model's capacity to elucidate the economic and spatial structure of urban agglomerations. The mathematical representation of the improved gravitational model can be expressed as:

$$ELink_{\underset{ij}{\rightarrow}} = \frac{X_{ij}}{X_{ij} + X_{ji}} \times \frac{TNLI_i \cdot TNLI_j}{D_{ij}{}^2} \tag{4}$$

where $ELink_{\underset{ij}{\rightarrow}}$ denotes the intensity of economic linkage between city $i$ and city $j$. $TNLI_i$ and $TNLI_j$ denote the total NTL indices of city $i$ and city $j$, respectively. $X_{ij}$ and $X_{ji}$ represent the trend of population movement from city $i$ to city $j$ and from city $j$ to city $i$, respectively. $D_{ij}$ indicates the economic distance between city $i$ and city $j$. Similarly, the intensity of economic linkage between city $j$ and city $i$ can be obtained as $Elink_{\underset{ji}{\rightarrow}}$.

Traffic Network Construction

For the Pearl River Delta and surrounding city clusters, inte-city transportation is primarily provided by buses, trains, high-speed trains, ships, and airplanes. Considering the limited availability of airports and public transportation facilities in the Pearl River Delta and surrounding cities during the research period, as well as the challenges in collecting historical data on bus services and specific service information for waterway passenger transportation, we opted to use train schedule data to represent the connectivity of intercity transportation. Train and high-speed train frequency data can, to some extent, reflect the strength of intercity transportation links. Accordingly, this study constructed the transportation network of the study area based on the frequency of train connections between cities. If direct train connections between two cities are unavailable, the intensity of transportation links between those cities is considered zero, and the intensity of transportation links between cities with direct train connections is based on the train frequencies. In this study, based on the previously researched traffic link strength [39], the trains were categorized as high-speed and normal-speed. Different weights were assigned to each type of train based on their respective speeds. The standard hourly speed of trains increases as transportation technology advances. In some years, high-speed trains between some cities were not opened. Therefore, this study calculated the average hourly speed of all high-speed trains that stopped in Guangzhou in 2020 as the reference standard hourly speed denoted by $V_{st}$. An improved mathematical formula for quantifying the intensity of traffic links is expressed as:

$$TLink_{\underset{ij}{\rightarrow}} = \frac{V_H}{V_{St}} \cdot P_H + \frac{V_N}{V_{St}} \cdot P_N \tag{5}$$

where $TLink_{\overrightarrow{ij}}$ denotes the intensity of traffic connection between city *i* and city *j*; $V_H$ and $V_N$ represent the average running speed of high-speed trains and the average running speed of general trains from city *i* to city *j*, respectively. $P_H$ and $P_N$ are the frequencies of high-speed trains and general speed trains from city *i* to city *j*, respectively.

Information Network Construction

With the advent of the information age, the flow of timely, convenient, and spatially unconstrained information enables frequent interactions among cities and plays an increasing role in promoting urban interaction and regional integration [50]. According to data provided by the "Statcounter" website (https://gs.statcounter.com/search-engine-market-share/all/china (accessed on 3 August 2022)), Baidu's search engine has an average market share of 72.1% in China between 2014 and 2020. Baidu is the largest search engine used by Internet users in China. Researchers have utilized the Baidu index to analyze information links between cities, including a study conducted by [51]. Compared with the search indices of other platforms, the Baidu search index strongly reflects the attention of domestic netizens to other cities. This study employed city names as search keywords and cities as the search area to obtain the data involving attention from city to city. Notably, the level of city interest is influenced by events of significant social impact in the region during a particular time frame. To account for the influence of unforeseen events, this study collected daily attention data between pairs of cities during the study period and calculated the mean value to determine the level of information attention between cities. To reflect the variability and directionality, the information linkage intensity between city *i* and *j* city is defined as $ILink_{\overrightarrow{ij}}$, and can be mathematically expressed as:

$$ILink_{\overrightarrow{ij}} = I_{ij} \qquad (6)$$

where $ILink_{\overrightarrow{ij}}$ denotes the annual average concern value of city *i* to city *j*.

Network Structure Analysis

The interactive communication between cities is unequal, and the inequality of intercity communication is described adequately via the directed weighted network. Therefore, this study adopted a directed weighted network to investigate the spatial structure and evolution of economic, information, and transportation networks in the PRD city agglomeration and surrounding cities. The variation between different networks in time series was described by improving the normalization of network links as follows:

$$Link_{nor(\overrightarrow{ij},t,k)} = \frac{Link_{(\overrightarrow{ij},t,k)}}{Max(Link_{(2014,k)}, Link_{(2017,k)}, Link_{(2020,k)})} \qquad (7)$$

where $Link_{nor(\overrightarrow{ij},t,k)}$ denotes the normalized k-functional linkage intensity from city *i* to city *j* in year *t*; $Link_{(\overrightarrow{ij},t,k)}$ indicates the to-be-normalized k-functional linkage intensity from city *i* to city *j* in year *t*, and $Max(Link_{(2014,k)}, Link_{(2017,k)}, Link_{(2020,k)})$ denotes the maximum values of k-functional linkage intensity in all study stages; k indicates the kind of functional network (one of the economic networks, transportation networks, or information networks).

The above functional network was also analyzed from the perspective of urban nodes and overall network structure using network science indicators.

### 3.2.2. Urban Node Analysis
Centrality

The centrality of a node can be used to determine the dependence of other nodes on the specified node, which is generally expressed using degree centrality, tightness, and intermediate centrality. Among different city clusters of functional networks, each city exhibits different intensities of functional linkages with other cities, resulting in different

weights. This study focuses on the degree of centrality (DC) and betweenness centrality (BC) of urban nodes.

Considering that the object of this study is a directed weighted network since the edges are assigned weights ($W_i$), the nodes connected to the edges also carry weights (point weights), denoted using ($S_i$).

$$S_i = \sum W_i \tag{8}$$

In a directed weighted network, the edge weights $W_i$ are directional, and the weights of nodes are divided into incoming weights $S_i^{in}$ and outgoing weights $S_i^{out}$ according to the pointing relationship of $W_i$. The mathematical expression is as follows:

$$S_i^{in} = \sum w_i^{in} \tag{9}$$

$$S_i^{out} = \sum W_i^{out} \tag{10}$$

The BC of a node is the sum of the weights of all edges connected to a city node in the city cluster network and is used to describe the ability of a city node to occupy resources in an urban group. According to Freeman's study [52], the mathematical expression of DC for node $i$ is as follows:

$$DC_i = S_i^{in} + S_i^{out} \tag{11}$$

where $DC_i$ denotes the degree of centrality of the city node $i$, and $S_i^{in}$ and $S_i^{out}$ represent the point-in and point-out rights of city node $i$. In the city cluster network, the higher the *DC* of a city node, the higher its possibility of becoming the network center.

The *BC* of a node is used to describe the ability of an urban node to control network resources. Freeman defines the *BC* of node $i$ as:

$$BC_i = \sum_{j<k} \frac{g_{jk(i)}}{g_{jk}} \tag{12}$$

where $g_{jk(i)}$ represents the number of times city $i$ is on the shortest path between city $j$ and city $k$ in the city cluster network other than city $i$. The term $g_{jk}$ represents the sum of all shortest paths between city nodes $j$ and $k$. For city cluster networks, the BC of a city indicates the ability of that city to control the connections between other cities and is the main indicator of the importance of city nodes in the city cluster network.

Nodal Symmetry

Nodal symmetry [53] is a concept used to assess the absorptive and radiative capacities of a city node within a city cluster network. It quantifies the balance between the node's ability to absorb resources from other nodes and its ability to radiate resources to other nodes. A higher nodal symmetry value indicates a stronger absorptive effect, suggesting that the node plays a central role in resource absorption within the city cluster network. On the other hand, a lower nodal symmetry value indicates a more pronounced radiative effect, indicating that the node primarily acts as a resource distributor within the network. The mathematical expression for nodal symmetry is as follows:

$$NS_i = \frac{S_i^{in} - S_i^{out}}{S_i^{in} + S_i^{out}} \tag{13}$$

where $NS_i$ denotes the node symmetry, which is used to quantify the balance of node interactions. $NS_i \in (-1,1)$, when node $i$ has no edge pointing to other nodes; $NS_i$ has the maximum value of 1 and the minimum value of $-1$.

Analysis of Overall Network Structure

To investigate the overall network structure of different functional networks and the evolution of network structure at different research stages, this study used graph entropy to determine the non-homogeneity of networks [54]. Before introducing graph entropy, we first defined the node's importance $I_i$.

$$I_i = s_i / \sum_{i=1}^{N} s_i \tag{14}$$

where $I_i$ is the node importance of the city node $i$; $N$ is the number of city nodes in the city cluster network; and $S_i$ denotes the total weight of the city node $i$. When $S_i = 0$, city node $i$ is not meaningful to our discussion at this point, we only considered the case of $S_i > 0$. At this point, the mathematical form of graph entropy $E$ can be expressed as:

$$E = -\sum_{i=1}^{N} I_i \times \ln I_i \tag{15}$$

when the network is perfectly uniform ($I_i = 1/N$), $E_{\max} = \ln N$; when the network is most inhomogeneous (a star network), $E_{\min} = [\ln 4(N-1)]/2$. To ensure that $E$ is not affected by the number of network nodes $N$, we normalized the graph entropy, which is expressed mathematically as follows:

$$E' = \frac{E - E_{\min}}{E_{\max} - E_{\min}} = \frac{-2 \sum_{i=1}^{N} I_i \ln I_i - \ln 4(N-1)}{\ln N^2 - \ln 4(N-1)} \tag{16}$$

where $E'$ denotes the standard structural entropy of the network, $E' \in [0, 1]$. In this study, graph entropy is employed to quantify the degree of developmental variability among cities within the city cluster network. A higher value of graph entropy signifies a reduced level of developmental imbalance among cities in the network. Conversely, a lower value of graph entropy suggests a greater degree of developmental imbalance among cities in the network.

## 4. Results

### 4.1. Network Space Distribution

Based on the data on night lighting, road networks, train stopping times, and the Baidu index, we calculated the intensities of economic, information, and traffic linkages in the PRD city cluster and its surrounding cities from 2014 to 2020 using Equations (4)–(6), respectively. To ensure comparability, the intensities of different functional linkages were normalized using Equation (7). The resulting normalized linkage intensities between city nodes were utilized to construct different functional networks. The networks and metrics were calculated using Python 3.9. The spatial distribution of the networks is visualized in Figure 3.

In general, the PRD city cluster and its surrounding cities exhibit a spatial pattern centered around Guangzhou, Foshan, and Shenzhen, with a gradual outward radiation based on factors such as economy, transportation, and information. As one moves away from the central cities, the edges of the cluster become sparser, and their weights diminish. This spatial distribution reflects the polarization effects observed in economic and transportation networks, whereas the information network displays a more uniform structure.

In the case of economic and transportation networks, the variations among cities are influenced by factors such as economic scale, population size, road network density, accessibility, and train schedules. Cities with larger economic scales, better road conditions, more frequent direct trains, and overall higher quality tend to have stronger intercity connections.

The flat structure observed in the information network can be attributed to the advancements in network technology, which have significantly reduced the economic cost associated with network utilization. The widespread use of the Baidu online search engine

has greatly facilitated information acquisition, diminishing the impact of geospatial location on the flow of information between cities.

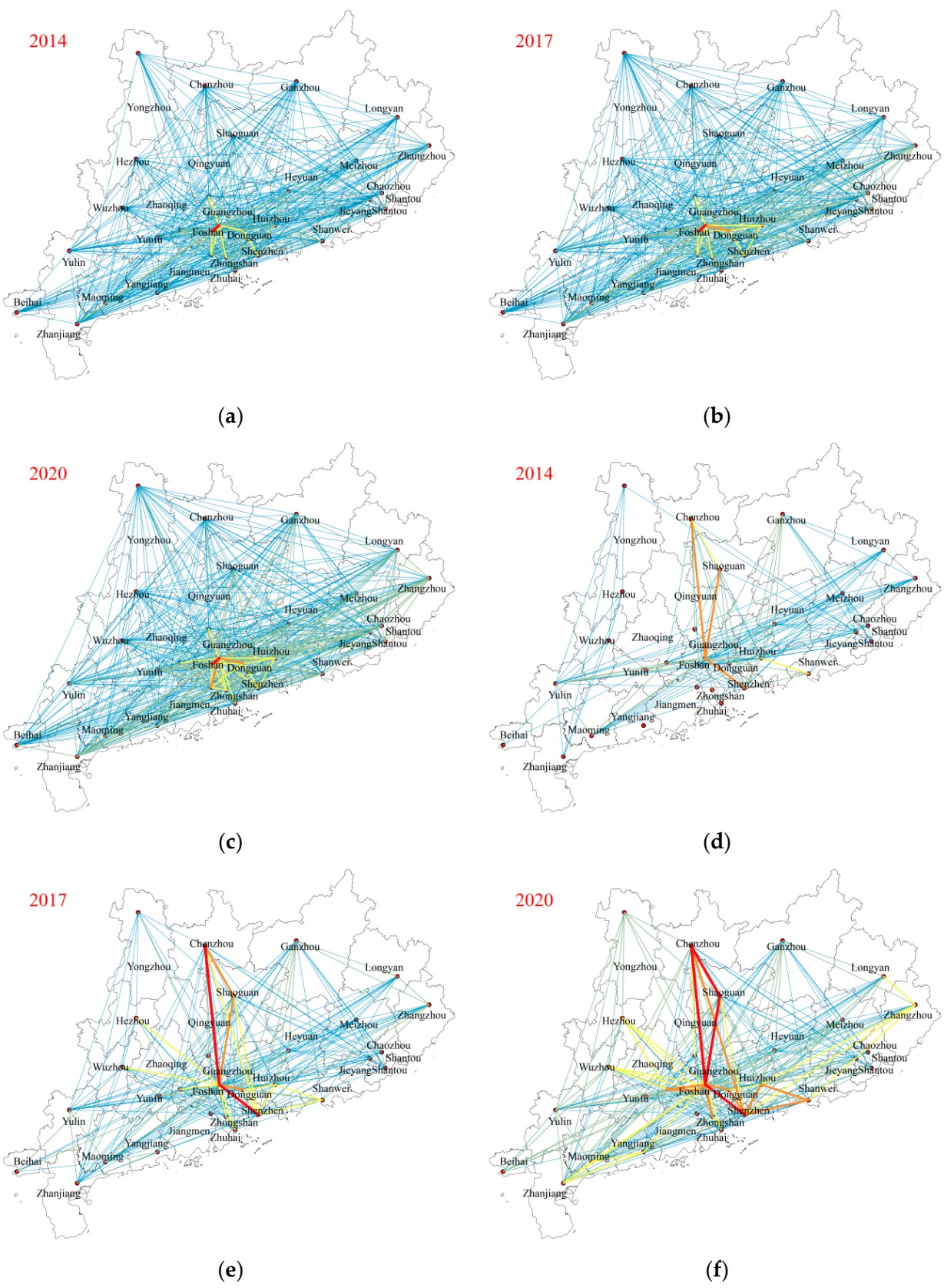

**Figure 3.** *Cont.*

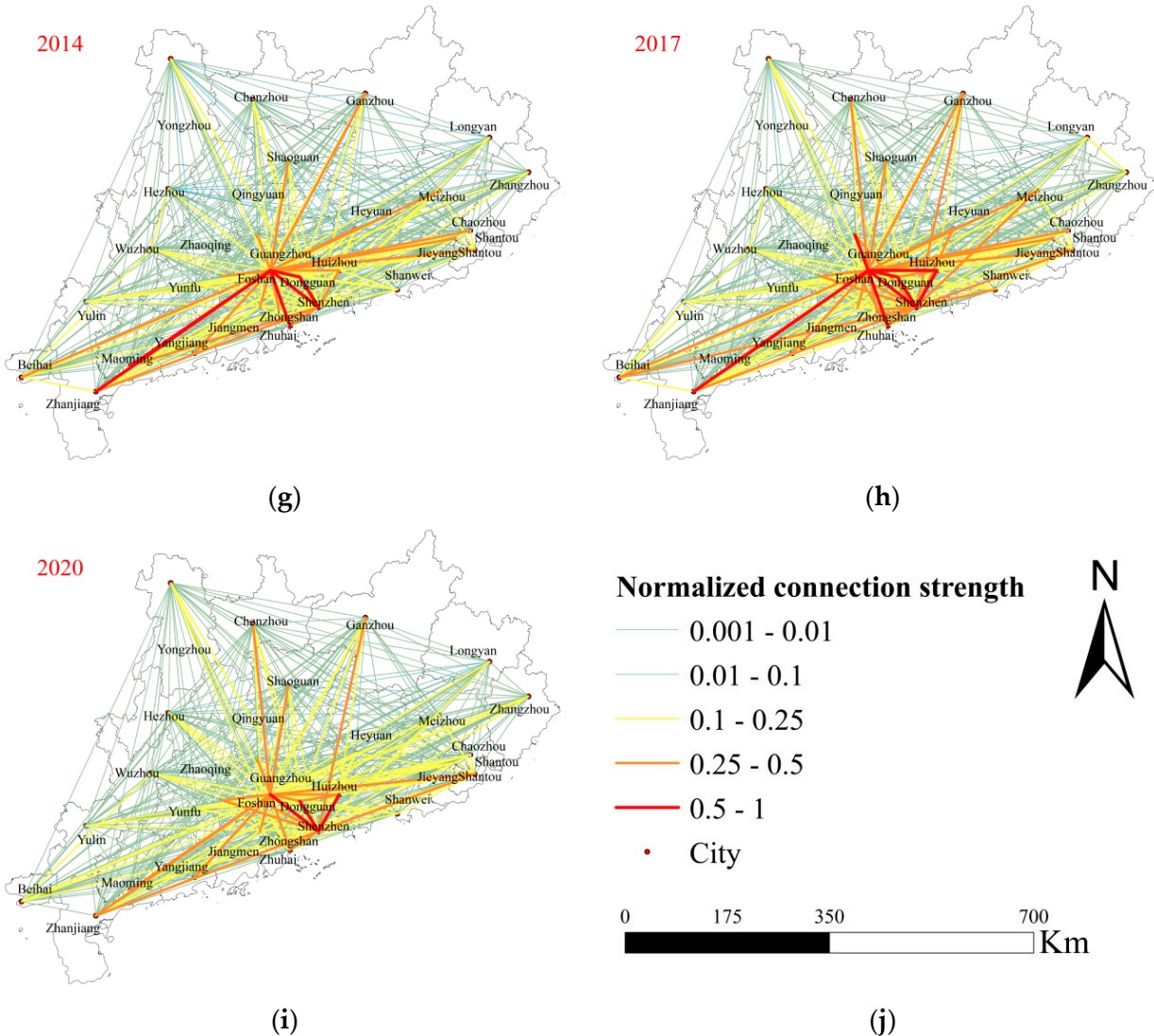

**Figure 3.** Spatial distribution of different functional networks: (**a**−**c**) Economic Network. (**d**−**f**) Transportation Network. (**g**−**i**) Information Network. (**j**) Legend.

From the perspective of network structure evolution, the transportation network transitioned from a single-axis structure to a "△" structure from 2014 to 2020. Guangzhou, Shenzhen, and Dongguan served as transportation centers and gradually extended their reach towards the east, west, and north of Guangdong province. As for the economic network, its structure evolution from 2014 to 2020 followed a similar pattern to the transportation network, but to a lesser degree. Initially, the economic network centered around Guangzhou and Foshan, but as the network structure evolved, Shenzhen and Dongguan gradually moved closer to the center of the economic network, ultimately forming an inverted "V" shaped economic network structure with Guangzhou, Foshan, Shenzhen, and Dongguan as the core cities. In terms of the information network, there were no significant changes in the overall structure from 2014 to 2020. All nine cities in the Pearl River Delta region held central positions in the information network, with Guangzhou consistently occupying a core position. However, over the course of the study period, the role of Guangzhou as an information center has weakened, and cities such as Shenzhen, Dongguan, Foshan, and Huizhou, which have shown strong development, have gained a higher status in the information network.

Overall, the spatial patterns and connectivity in the PRD city cluster and its surrounding cities are shaped by the interplay of economic, transportation, and information factors, with each network exhibiting distinct characteristics.

### 4.2. City Node Analysis

Based on the spatial distribution of different networks, we further analyzed the centrality of each city in the city cluster. Figure 4 shows the joint distribution of degree centrality (DC) and betweenness centrality (BC) of each city in different networks in the PRD urban agglomeration and its surrounding cities at different stages.

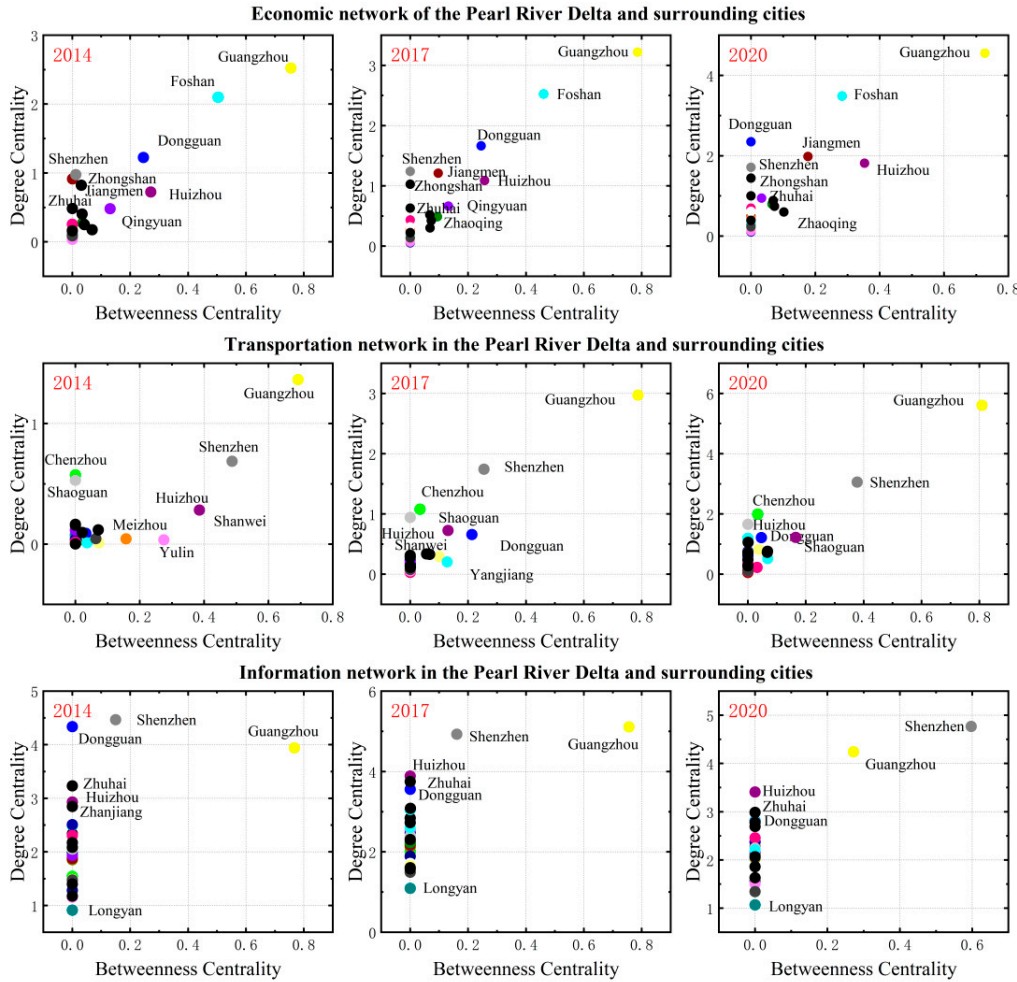

**Figure 4.** Joint distribution of intermediate centrality and degree centrality.

Figure 4 reveals a significant correlation between *DC* and *BC* in the economic and transportation networks of the PRD city cluster and its surrounding cities, with cities with higher *DC* also exhibiting higher *BC*. This suggests that the resource level of a city within an urban agglomeration determines its control over resource exchange among different cities. In various functional networks, a city with a high *DC* represents a direct functional connection with other cities, while cities with a high *BC* act as crucial "bridges" in the urban network, exerting strong control over resources.

However, it is worth noting that the information network formed by the Pearl River Delta city cluster and surrounding cities does not exhibit a clear correlation between *BC* and *DC*. The main reason for this could be that the interaction patterns of information elements are constantly evolving and being updated with the advancement of information technology and internet communication technology, to the point where they are no longer significantly constrained by geographical distance factors. The interaction processes among

the cities in the research area in the information space do not demonstrate a "bridge" role but rather direct interactions between the entities themselves.

Based on the results, we found that Guangzhou exhibits a dominant position in all three functional networks. Foshan ranks second in terms of importance in the economic network. Shenzhen ranks second in terms of importance in transportation and information networks. Considering that the formation of urban agglomerations and urban networks serves economic development, this study identifies Guangzhou and Foshan as the core city nodes of the urban network. Dongguan, Jiangmen, Huizhou, Shenzhen, and Zhongshan are considered secondary core city nodes. Other cities are classified as peripheral city nodes. The spatial structure of the Pearl River Delta urban agglomeration and surrounding cities forms a three-tier hierarchical structure. Next, we visualize the total city connectivity of different cities in different functional networks, as shown in Figures 5–7.

The economic network analysis (Figure 5) reveals that the total economic linkages between core cities and surrounding cities constituted 34.4%, 31.2%, and 28.5% of the total economic linkages within the study area in 2014, 2017, and 2020, respectively. Similarly, the total economic linkages between sub-core cities and surrounding cities represented 34.7%, 33.9%, and 33% of the total economic linkages within the study area in 2014, 2017, and 2020, respectively. The analysis of the proportion of total economic ties between core cities and sub-core cities revealed a distinct circular pattern in the PRD and its surrounding cities, indicating notable developmental disparities between cities at different levels.

We conducted an analysis of low-economic-density areas and based on the distribution of the total economic connections, we found that cities such as Shanwei, Jieyang, Shantou, Chaozhou, Zhangzhou, and Ganzhou have a certain share in the total economic connections of the entire Pearl River Delta city cluster and its surrounding cities. Among them, Zhangzhou, Shantou, and Ganzhou are cities within the Cross-Strait Economic Zone, with Zhangzhou being one of the key cities in the Economic Zone on the West Side of the Straits. The economic interaction between the Economic Zone on the West Side of the Straits and the Pearl River Delta city cluster also reflects this to some extent.

As for cities like Chenzhou, Yongzhou, Hezhou, Wuzhou, Yulin, and Beihai, we found that these cities had a very low share of total economic connections in different stages of the study. For example, Wuzhou, as an important node city in the Pearl River-Western River Economic Belt, has a relatively low share of connections, indicating that there is room for improvement in the implementation of the Pearl River-Western River Economic Belt Development Plan. Yulin, as a crucial gateway and key node city for connecting the Pearl River Delta city cluster and the Beibu Gulf Economic Zone, and Beihai, as an important node city in the Beibu Gulf city cluster, have low shares of economic connections, highlighting the need to strengthen economic ties between the Pearl River Delta city cluster and the Beibu Gulf city cluster.

In the transportation network (Figure 6), the total transportation links between core cities and surrounding cities accounted for 28.1%, 25.6%, and 26.4% of the total transportation links within the study area in 2014, 2017, and 2020, respectively. The total transportation links between sub-core cities and surrounding cities accounted for 24.1%, 24.4%, and 24.4% of the total transportation links within the study area in 2014, 2017, and 2020, respectively. The share of total connections between core cities and surrounding cities in the transportation network was lower than in the economic network, partly because Foshan is a core city. Its geographical location is too close to the main core of Guangzhou. Assuming the number of transportation connections without direct trains between cities is zero, zero transportation connections were found between Foshan and some cities using Guangzhou as a transit. However, Foshan's position in the PRD city cluster cannot be denied. In terms of the ratio of traffic connections between core and sub-core cities to total regional traffic connections in different years, the difference in traffic resources between different cities in the region has declined.

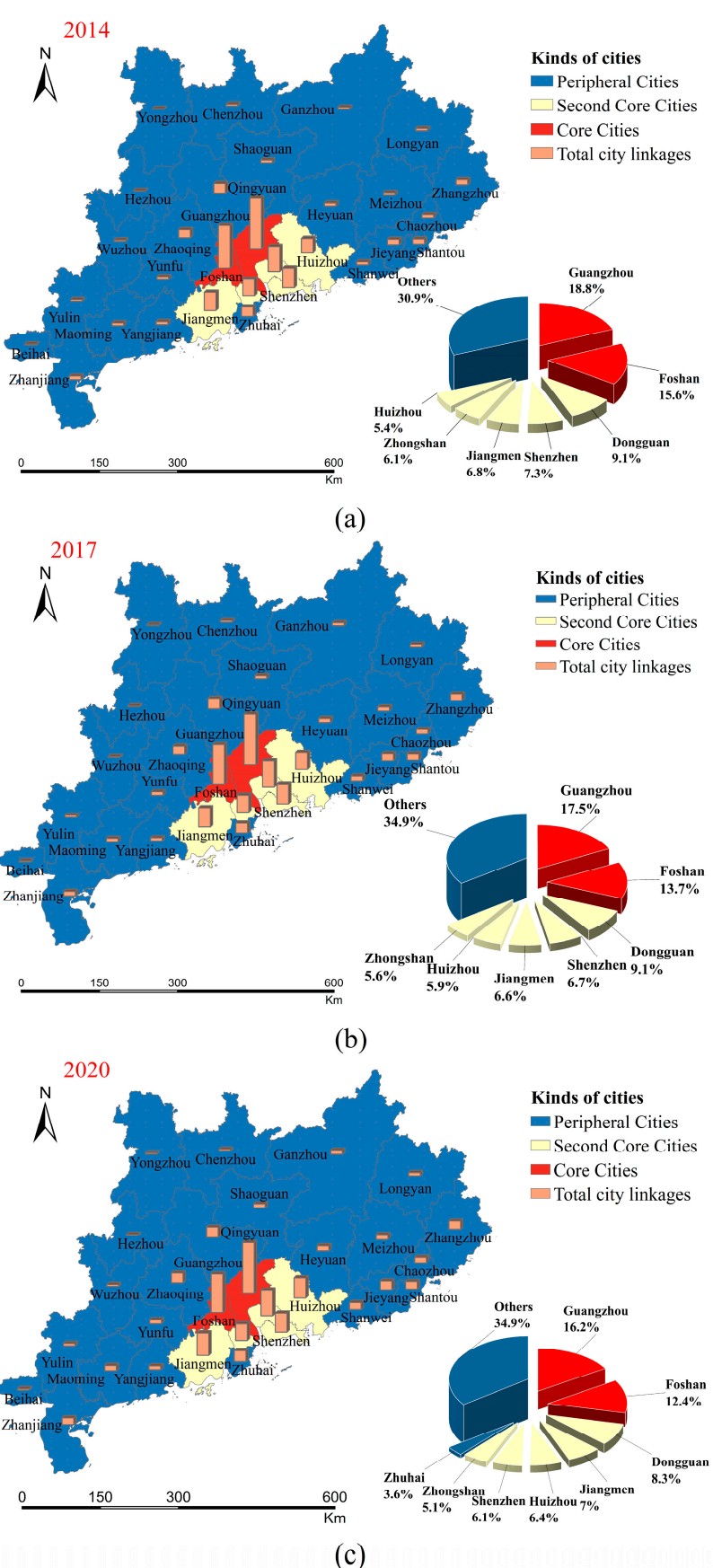

**Figure 5.** The total number of connections between different cities in the economic network: (**a**) Economic Network in 2014. (**b**) Economic Network in 2017. (**c**) Economic Network in 2020.

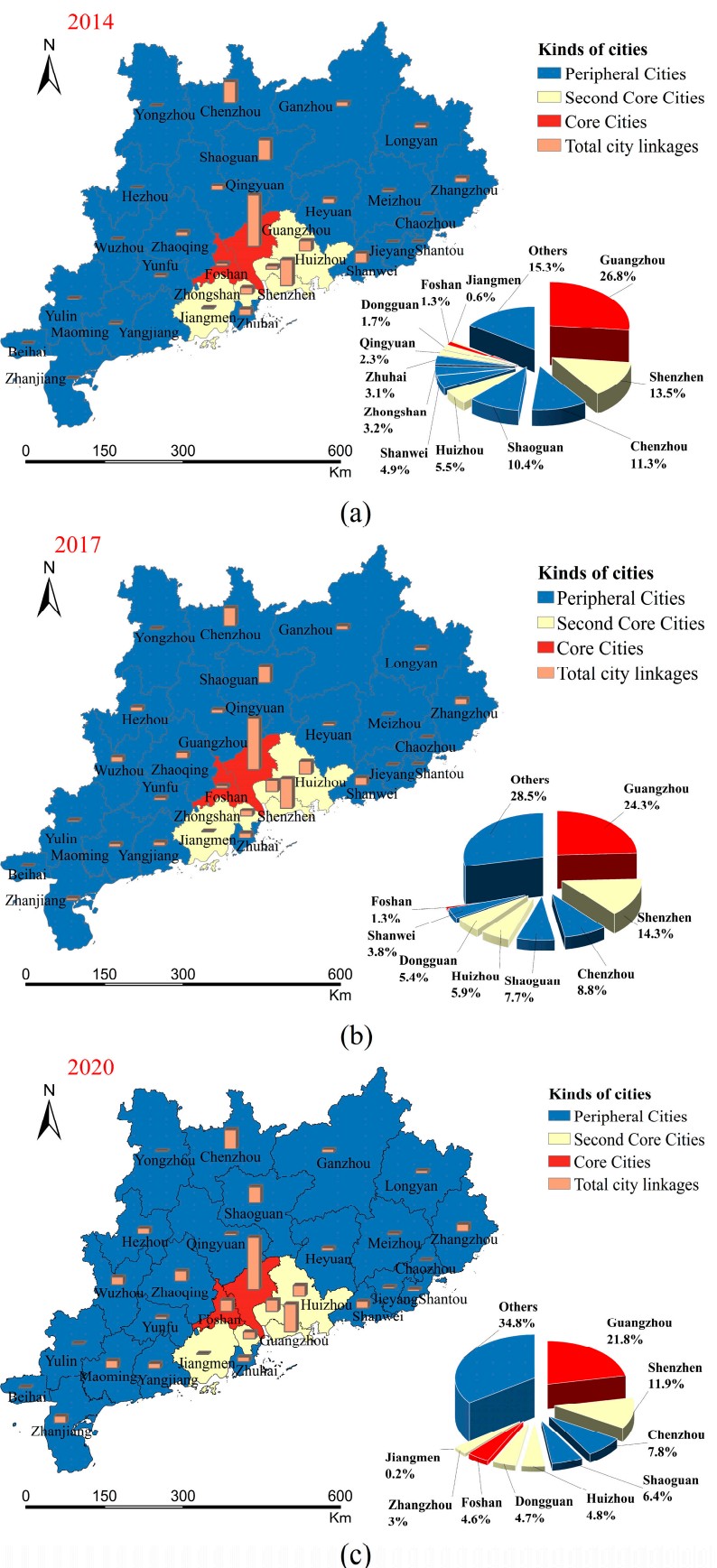

**Figure 6.** The total number of connections between different cities in the transport network: (**a**) Transport Network in 2014. (**b**) Transport Network in 2017. (**c**) Transport Network in 2020.

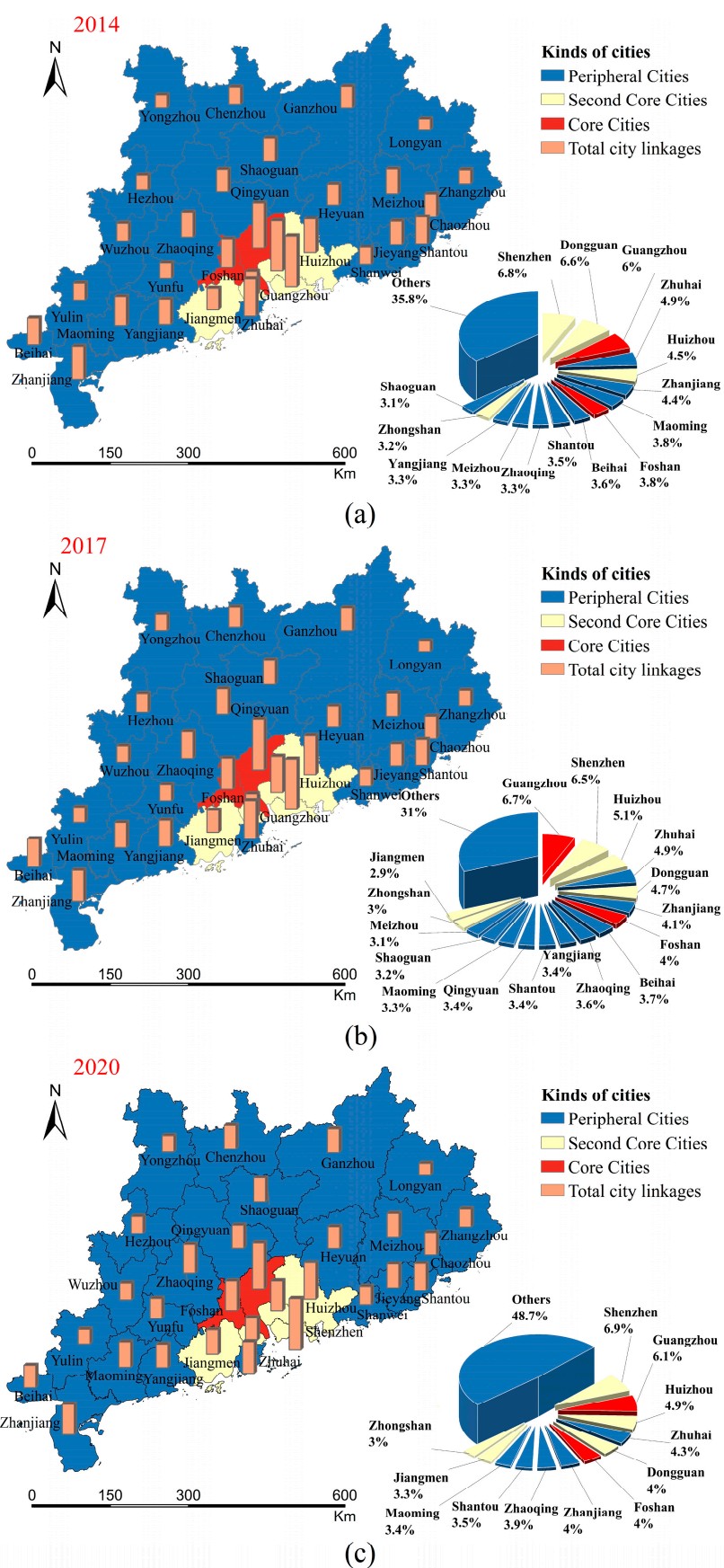

**Figure 7.** The total number of connections between different cities in the information network: (**a**) Information Network in 2014. (**b**) Information Network in 2017. (**c**) Information Network in 2020.

We conducted an analysis of low-traffic-density areas, and based on the distribution of total traffic connections, we found that cities such as Chenzhou, Shaoguan, and Shanwei had a relatively high share of total traffic connections in different stages of the study. This indicates that the core of the transportation network mainly lies in the direction of the northern part of Guangdong Province, serving as an important corridor connecting the Pearl River Delta city cluster with inland cities in terms of transportation.

Furthermore, when comparing different stages of the study, we observed that cities such as Wuzhou, Hezhou, Zhangzhou, Zhanjiang, and Maoming have seen an increase in their share of total traffic connections. This means that the government has made adjustments to the railway network and train frequencies in different stages of development to enhance the transportation connections between the Pearl River Delta city cluster and inland cities as well as the surrounding economic areas.

In the information network (Figure 7), we observe that the percentage of information links between core cities and surrounding cities in the study area is 9.8%, 10.8%, and 10.1% in 2014, 2017, and 2020, respectively. Similarly, the percentage of information links between sub-core cities and surrounding cities was 24%, 23.2%, and 22% in the respective years. The proportion of information links between core cities and surrounding cities is much lower compared to their proportions in the economic and transportation networks.

For edge cities such as Yongzhou, Chenzhou, Zhangzhou, Zhanjiang, Beihai, and Maoming, the share of their information connections remains relatively stable across different study periods. This suggests that with the rapid development of internet technology and information transmission channels, as well as the weakening impact of geographical distance and transportation convenience on intercity information exchange, the absolute differences in information connections between cities within the study area are not significant.

In summary, the results indicate that the construction and development of the information network have helped mitigate the limitations imposed by geographic distances, enabling more balanced and widespread information exchange among cities in the region.

### 4.3. Comparative Analysis of Urban Absorption and Expansion

After conducting an analysis of the centrality of cities with distinct functions at various stages, we identified the core cities within the urban agglomerations. Subsequently, we proceeded to examine the specific resource flows within each city during different stages of the urban agglomeration. As shown in Figure 8.

According to the principle of small intra-group and large inter-group variations [55], nodal symmetry values exceeding 0.15 were considered indicative of a substantial net inflow of urban resources. Nodal symmetry values greater than 0 but less than 0.15 were classified as representing a minor net inflow of urban resources. Conversely, nodal symmetry values greater than −0.15 but less than 0 indicated a minor net outflow of urban resources, while values lower than −0.15 were considered indicative of a significant net outflow of urban resources.

Based on the analysis of urban centrality, it was observed that Guangzhou, as the core city, along with Shenzhen and Dongguan, as secondary core cities, experience significant net inflows of urban resources in both the economic and information networks. In the economic network, this can be attributed to their higher economic levels within the region and comparatively better living standards, which attract a considerable influx of labor, businesses, and developmental resources from the surrounding cities. As a result, these conditions create a favorable environment for economic development, reinforcing their core positions in the economic network. Furthermore, despite Foshan's proximity to Guangzhou and its excellent transportation conditions, it experiences a slight net inflow of urban resources and remains in the secondary core of the economic network.

In the transportation network, the flow of urban transportation resources is greatly influenced by the total number of train services at the city's railway stations. In cities with a low number of trains stops, the accuracy of the results obtained through nodal

symmetry analysis, which determines the net inflow and outflow of urban resources, is compromised. Consequently, it becomes challenging to accurately depict the inflow and outflow of resources in these cities.

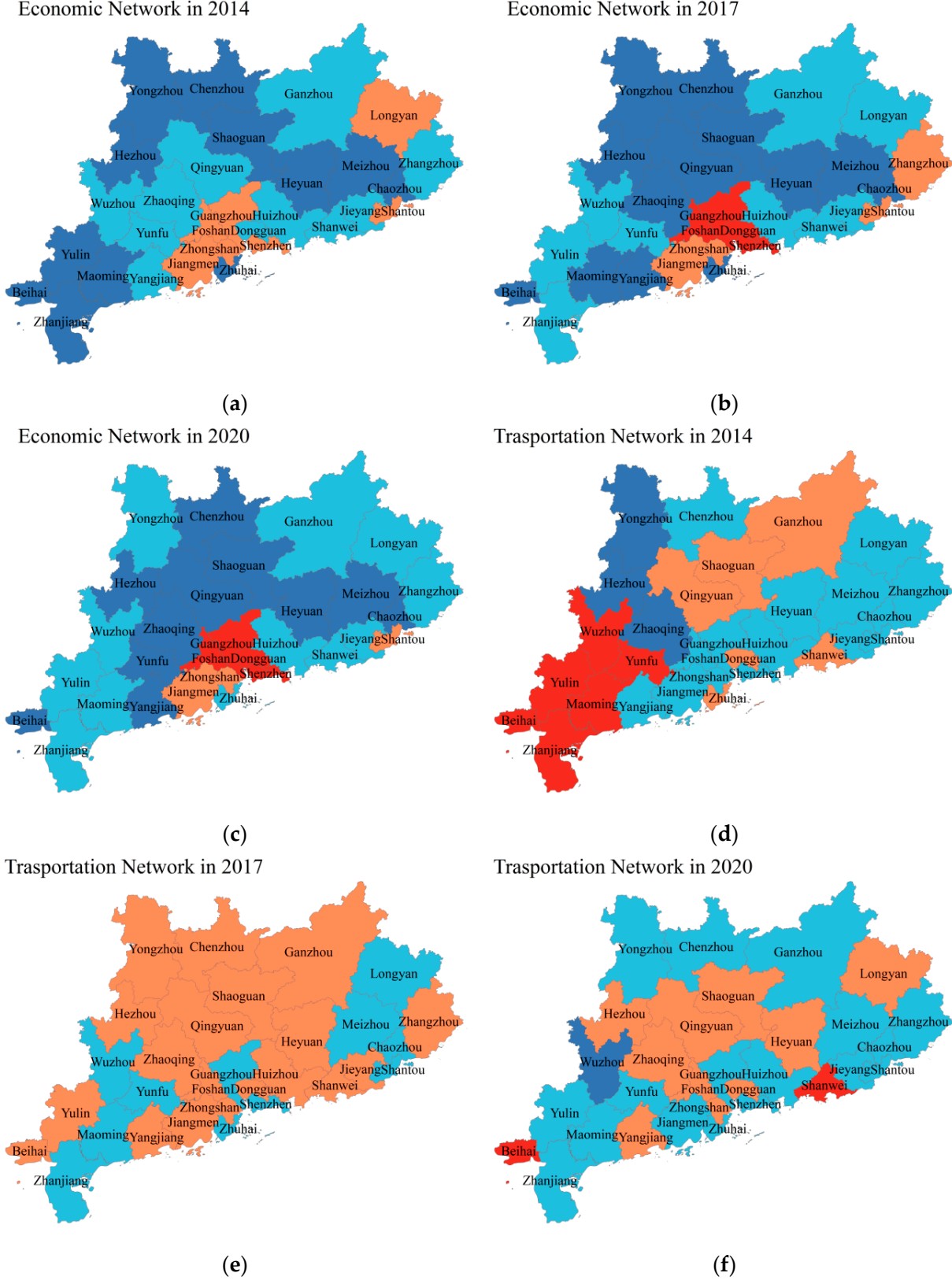

**Figure 8.** *Cont.*

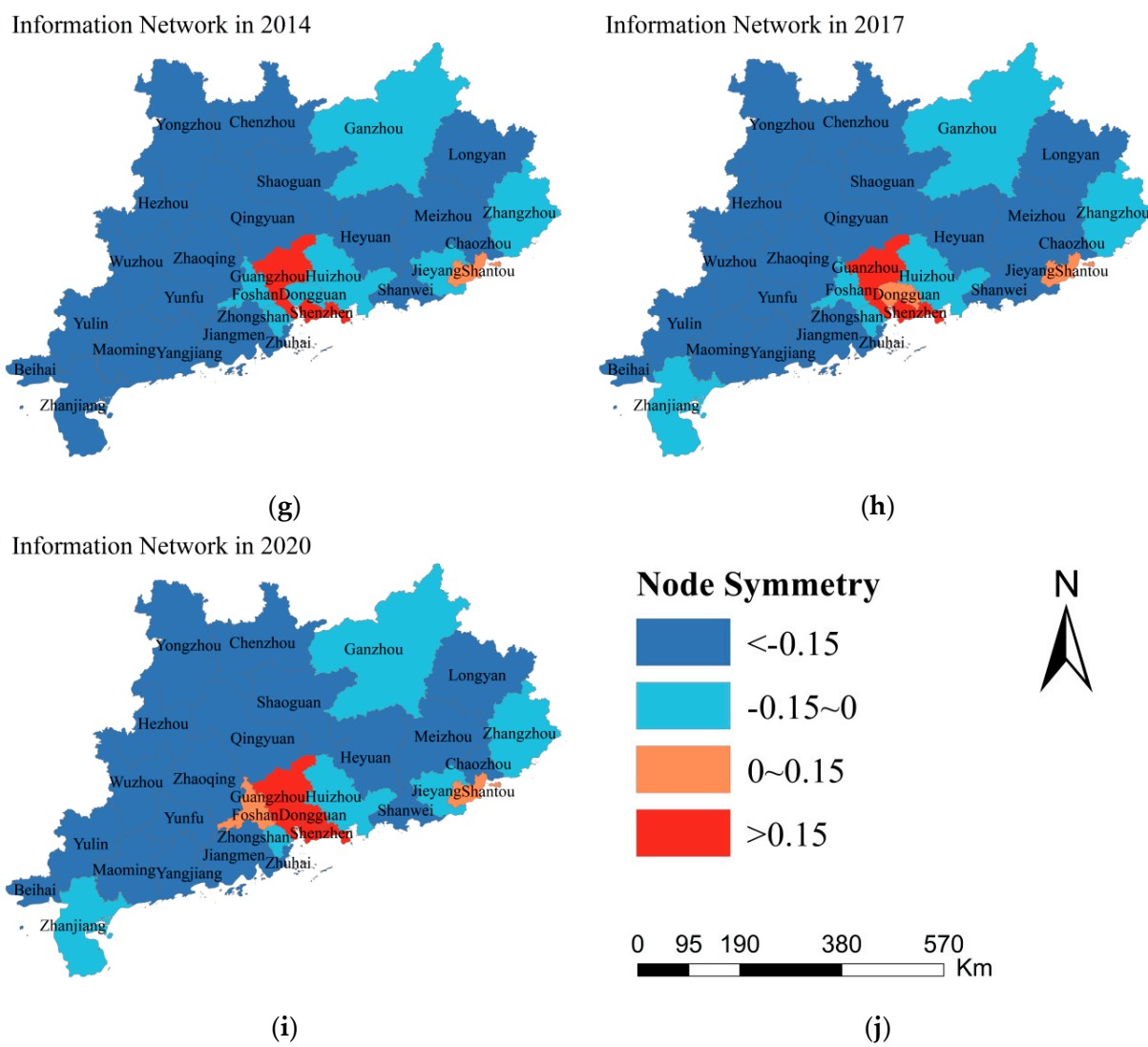

**Figure 8.** Distribution of resource inflows and outflows to and from urban nodes of different functional networks in the PRD city cluster and surrounding cities: (**a**−**c**) Economic Network. (**d**−**f**) Transportation Network. (**g**−**i**) Information Network. (**j**) Legend.

Regarding the information network, the core cities' relatively high levels of comprehensive development at different stages in the region, along with their larger economic and population sizes compared to other cities, attract more attention from the surrounding cities. These surrounding cities prioritize economic growth and population size, resulting in a dominant central position for the core cities in the information network. This underscores the significance of factors such as comprehensive development and reputation in shaping urban development.

### 4.4. Overall Network Structure

The nodal symmetry analysis provides a clearer insight into the function of cities in the region. To analyze the degree of imbalance in urban development across different functional networks and the evolution of the network structure, we used the standard structural entropy of the network and the growth rate of entropy in the network structure, represented by the change in the network entropy at the same time.

As shown in Table 3, the standard structural entropy of different functional networks in all three stages showed the following pattern: information network > economic network > transportation network. Thus, the information network comprising the PRD urban

agglomeration and its surrounding cities was less unbalanced than the urban economic and transportation networks. The information network is based on the keyword searches of Internet users using online search engines. The differences in geospatial location and transportation conditions have a diminished impact on the exchange of information between different cities, thus contributing to the basic pattern of the information network. The economic network is more polarized in terms of urban nodes compared with the transportation network, while the economic network is less unbalanced.

**Table 3.** Standard graph entropy of different functional networks in the Pearl River Delta city cluster and surrounding cities.

| | **2014** | **2017** | **2020** |
| --- | --- | --- | --- |
| Economic Network | 0.404 | 0.493 | 0.565 |
| Transportation Network | 0.221 | 0.345 | 0.476 |
| Information Network | 0.855 | 0.866 | 0.899 |

Accordingly, we computed the graph entropy growth rate for various networks. Table 4 presents the evolution of different networks, which can be characterized by the graph entropy growth rate within their respective network structures. The growth rate of graph entropy in the economic and transportation networks exhibited a decreasing trend over time, indicating the maturity and stability of the network structure as it developed. Conversely, the graph entropy growth rate increased during the progression of the information network, suggesting structural instability within the information network. This phenomenon may be attributed to the attainment of a more balanced state among cities in the region within the information network, resulting in reduced polarization. Consequently, the presence of information gaps in the region contributed to the instability observed in the information networks.

**Table 4.** The growth rate of graph entropy of different functional network.

| | **2014–2017** | **2017–2020** | **2014–2020** |
| --- | --- | --- | --- |
| Economic Network | 22.0% | 14.6% | 39.86% |
| Transportation Network | 56.1% | 38.0% | 115.4% |
| Information Network | 1.3% | 3.8% | 5.1% |

## 5. Discussion

This study examines the intercity linkages within the PRD city cluster and surrounding cities from three functional perspectives: economic flows, traffic flows, and information flows. It further investigates the evolution of different functional network structures. The study makes the following key contributions:

(1) The study reveals variations in the spatial distribution patterns of multidimensional functional networks, which highlight the functional imbalances between the PRD city cluster and the surrounding cities. These findings provide valuable insights for policymakers and urban planners, enabling them to comprehend the dynamics of intercity relations and formulate targeted strategies for regional development.

(2) The identification of the core-sub-core-edge circle structure indicates the existence of a relatively stable spatial configuration within the PRD urban agglomeration and surrounding cities. This phenomenon implies that the influence of core cities on edge cities is limited. Understanding the core-periphery dynamics of the region is crucial for effective resource allocation, development strategy formulation, and regional cooperation initiatives.

However, there are some shortcomings in the study's results.

First, the suitability of the gravity model for analyzing spatial interactions in modern societies was not specifically validated. While Sun [56] utilized the gravity model

to construct an economic network of major urban agglomerations in China and found correlations between the model and empirical data, further empirical studies are required to demonstrate the applicability of gravity models in describing modern socio-spatial interactions. Although some studies [35,57–59] have utilized gravity models to describe socio-spatial interactions, additional empirical research is needed to strengthen the validity of this approach. This is an area that requires further improvement in future studies.

Second, our measurement of urban transport links using the frequency of direct trains between cities may not accurately represent the actual passenger and logistics flows between cities. Therefore, it may not serve as an accurate indicator of the intensity of transport links. Unfortunately, due to the unavailability of publicly accessible data sources that provide historical urban flows, real intercity movements of people and goods, and migration data, we are unable to verify the accuracy of the modeled traffic flow formulas. However, Jiang et al. [29] validated the usability of a similar method in the middle Yangtze River urban agglomeration. Hence, it is reasonable to employ this method in the PRD city cluster and surrounding cities as well.

Third, the city economic distance based on the "Transportation Accessibility Index" we constructed is only applicable to the road network conditions of the Pearl River Delta and its surrounding cities. This limitation prevents us from guaranteeing the applicability of the gravity model constructed in this study for assessing the intensity of economic connections in all regions. The accuracy of quantifying economic connections in other regions using this model remains to be examined. Improving the model to enhance the accuracy of evaluating the economic connectivity between cities and ensuring its universality will be the main focus of our future work.

In contrast to Peng's [35] study, our use of the gravity model incorporates the Night-Light Index to characterize the economic quality of cities instead of relying solely on traditional socio-economic data. This approach offers a more objective and accurate depiction of the urban economy. Furthermore, compared to the studies conducted by Jiang [29] and Sun [39], we adopt a more multidimensional perspective to examine the functional linkages among cities. This enables us to draw more comprehensive conclusions regarding intercity relationships. Additionally, our analysis encompasses the overall evolutionary trends of various functional network structures within the PRD city cluster and surrounding cities across different research stages. This provides a more in-depth understanding of the spatial structure of the city cluster.

Lastly, we would like to provide policy suggestions for urban departments and practitioners. At the regional scale, it is crucial to prioritize the transportation and information connections between different cities within urban agglomerations, considering the advancements in transportation and information technologies. Our analysis of spatial connections suggests that these connections can be adjusted based on regional economic development to optimize the spatial structure of urban agglomerations.

At the local scale, strengthening the radiation effect of core and sub-core cities on surrounding cities is essential. The limited radiation effect of core cities on peripheral cities, as observed in the spatial distribution of urban networks at different stages, indicates a polarization trend in the development of cities within the Pearl River Delta urban agglomeration and surrounding areas. Cities located at the regional periphery should actively enhance their connections with cities across the region and actively integrate into the development of the Pearl River Delta.

Gradually reducing administrative barriers is a crucial step. Although there are evident administrative barriers in economic and transportation networks, the flow of information between cities is gradually breaking down these barriers. Therefore, relevant departments in the Pearl River Delta and surrounding cities should leverage the advantage of information flow to overcome administrative and geographical restrictions. This will facilitate and guide cities to enhance economic, political, and cultural exchanges and cooperation, ultimately establishing a unified market. Overcoming development barriers in urban communication and cooperation can be achieved through these efforts.

## 6. Conclusions

This study utilized multiple data sources, including nighttime light data, Baidu search index data, and train stop schedule data, to construct functional networks of the Pearl River Delta urban agglomeration and surrounding cities from 2014 to 2020. The analysis employed network science metrics and complex network analysis methods to examine urban centrality, resource symmetry, and the evolution of network structures at both regional and local scales. The main findings of the study are as follows:

(1) The spatial distribution characteristics of the three functional networks (economic, transportation, and information) exhibit significant differences. Guangzhou and Foshan play central roles in the economic network, while cities like Chenzhou, Shaoguan, Guangzhou, Shenzhen, and Huizhou dominate the transportation network. The proportion of transportation connections in cities such as Shanwei, Dongguan, Foshan, and Zhongshan is gradually increasing over time. The interactions between the Pearl River Delta cities and surrounding cities are primarily concentrated in the northern part of Guangdong Province in terms of transportation. The information network displays a hierarchical structure, with the Pearl River Delta cities occupying a leading position, likely influenced by their comprehensive development strengths.

(2) The spatial structure of the Pearl River Delta urban agglomeration and surrounding cities can be divided into a three-tier hierarchical structure. Guangzhou and Foshan act as core cities, while Dongguan, Shenzhen, Huizhou, Zhongshan, and Jiangmen serve as secondary core cities. Other cities are considered peripheral cities. However, there have been no significant changes in the spatial structure over time, indicating a relatively weak radiation effect of core cities on peripheral cities.

(3) Core cities in the Pearl River Delta urban agglomeration and surrounding cities experience a net inflow of resources, while peripheral cities predominantly witness a net outflow of resources. This suggests that resource interaction between the Pearl River Delta cities and inland cities primarily results in resource aggregation. Enhancing the radiation power of core cities requires strengthening resource sharing, deepening economic cooperation, and increasing policy support.

(4) From 2014 to 2020, the economic network evolved from a uniaxial structure to an "inverted V" structure. The transportation network evolved from a uniaxial structure to a "△" structure. The information network did not show any obvious structural changes during its development, except for a star-shaped radial structure.

**Author Contributions:** Methodology, H.L. and S.N.; formal analysis, S.N.; writing—original draft preparation, S.N.; writing—review and editing, H.L. and S.N.; visualization, S.N.; supervision, H.L.; project administration, H.L.; funding acquisition, H.L. All authors have read and agreed to the published version of the manuscript.

**Funding:** The research was funded by the Jiangxi Social Science Foundation (21YJ23D) and the College-Industry Collaborative Breeding Project (202102245015).

**Institutional Review Board Statement:** Not applicable.

**Informed Consent Statement:** Not applicable.

**Data Availability Statement:** The data presented in this study are available on request from the corresponding author.

**Acknowledgments:** We would like to thank Piesat Information Technology Co., Ltd. for providing us with a platform for development. We are solely responsible for the opinions expressed in this study.

**Conflicts of Interest:** The authors declare no conflict of interest.

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
