# Peer review of "Analysis of Construction Networks and Structural Characteristics of Pearl River Delta and Surrounding Cities Based on Multiple Connections"

_sustainability, doi:10.3390/su151410917_

Round 1

Reviewer 1 Report

The author constructed a directed weighted network from the perspectives of economy, information and transportation, and discusses the inter-city connection between the Pearl River Delta urban agglomeration and inland cities and the evolution of the network spatial structure by using traditional statistical data, NPP/VIIRS NTL data, urban road network data, Baidu index, train stops and other data. This is an interesting research, but I still have some suggestions for this paper.

1. Line 310, , not reflected in formula (7).

2. There is no time annotation in Figure 3.

3. Tables 3, 4, and 5 consider visualizing through maps to make readers' reading more intuitive.

4. It is recommended to adjust the color scheme of the map in the text, highlight the main information, and increase the readability of the map while considering its aesthetics.

5. The language of the article needs further polishing, such as word errors, syntax error, etc.

The language of the article needs further polishing, such as word errors, syntax error, etc.

Author Response

Dear Reviewer,

Thank you for reviewing our manuscript and providing valuable feedback. We appreciate your comments and address them in the attached document.

We sincerely appreciate your review and constructive comments. We will diligently address each point you raised and make the necessary improvements to enhance the quality of the manuscript.

Thank you once again for your valuable input.

Best regards,
Shengdong Nie

Reviewer 2 Report

Detailed comments on linguistic and editorial errors as well as ambiguities in some wording are included in the attached PDF file (comments in the text). They cover about 30% of the volume of the reviewed article. When I realized the basic doubts about the methodology used, I stopped further checking the text.

1. The Zipf "gravity" model used by the authors is not suitable for the analysis of spatial interactions occurring in modern societies. It is a simple analogy to Newton's model - freely modified by the Authors. Before undertaking this research, the authors should thoroughly familiarize themselves with the contemporary and extensive literature on the application of models of spatial interactions in economics, for example, with the works of J.E. Anderson, P. Nijkamp, F. Lukermann and P.W. Porter and many others (bibliographical notes of some important items are included below).

2. In their research, the authors introduced the "corrected" economic distance (formula 2). Why is this distance called "economic distance"? There is no economic variable in this definition, for example, "transport costs" or "connection costs", etc. Moreover, what axioms does the distance (metric) thus defined satisfy? Is it true for example "triangle inequality"?

3. Since the authors study the structure of network systems (graphs), they should use the term "graph entropy".

Based on the reviewed article, it can be assumed that the authors undertook the research methodologically unprepared. This is due to their little orientation in the contemporary scientific literature on the issues of spatial interactions. If, despite my opinion, the work is published, a solid linguistic and editorial proofreading is also desirable.

Anderson J.E. (1979), A Theoretical Foundation for the Gravity Equation. The American Economic Review, 69 (1)

Anderson J.E. (2010) The Gravity Model. NBER working paper, 16576

Anderson J.E. Yotov Y.V. (2008) Gold Standard Gravity. NBER working paper, 17825

Chaney T. (2013) The Gravity Equation in International Trade: en Explanation. NBER working paper, 19285

Dzięcielski M, Kourtit K, Nijkamp P, Ratajczak W. (2021) Basins of attraction around large cities - A study of urban interactions spaces in Europe. CITIES, 119, 1-15.

Fotheringham A.S., O'Kelly M.E. (1989) Spatial Interaction Models: Formulations and Applications. Kluwer, Dordrecht

Kinces A., Nogy Z., Tóth G. (2015) Modelling the spatial structure of Europe. MPRA Paper no. 73957

Lukermann F., Porter P.W. (1960) Gravity and Potential Models in Economic Geography. Annals of the Association of American Geographies, 50 (4)

Nijkamp P., Ratajczak W. (2020) Gravitional Analysis in Regional Sciences and Spatial Economics: A vector Gradient Approach to Trade. International Regional Science Review, 44 (3-4), 400-431.

Author Response

(The authors gave the same response as above.)

Reviewer 3 Report

1. The literature review part is weak, with few study analyses on urban networks and structures, and more analysis based on the Pearl River Delta urban agglomeration, which is relatively one-sided. It is recommended to reorganize the literature review part.

2. The study references are too much focused on the Chinese region and should have an international perspective.

3. Although the authors use a lot of data and methods, the use of methods and data are not strongly correlated. It is suggested that the authors add a flowchart to make it clear how the data, methods, and research content correspond to each other.

4. The authors' core content analyzes four parts, Network Space Distribution, City Node Analysis, Comparative analysis of urban absorption and expansion and Overall network structure. In my opinion, the analysis of the study results is complicated, in other words, the authors write too many things, and it is suggested that the results part should be appropriately streamlined.

5. The discussion part should compare their study with other studies, rather than going for a repetition of the findings at the beginning.

6. The authors should elaborate on the contribution of the existence of this study.

7. The conclusion part is actually a repetitive presentation of the four studies, and it is recommended to streamline it and state the important conclusions and implications of this study.

Moderate editing of English language

Author Response

(The authors gave the same response as above.)

Round 2

Reviewer 2 Report

The responses to my comments in the previous review were polite but mostly evasive. The authors also did not pay attention to what I wrote with the text of the article in the attached PDF file. Most of them were comments on linguistic and editorial errors, but also quite significant doubts related to the methodology used. For this reason, this time I have included all important notes in the attached PDF file. Maybe now the authors will notice them.

Author Response

(The authors gave the same response as above.)

Reviewer 3 Report

Accept in present form

Author Response

Dear Reviewer,

Thank you very much for reviewing our paper and providing valuable feedback. We are delighted to hear that you recommend accepting our paper and find it acceptable in its present form.

Your expert opinion holds significant value for our research work and provides us with directions for further improvement and refinement. We will carefully consider your suggestions and incorporate appropriate modifications and revisions in the final version to better meet the requirements of the academic journal and readers' expectations.

Once again, we sincerely appreciate your review and recognition, and we are grateful for your valuable contribution to the publication process of our paper. We will submit the revised final version as soon as possible and hope to have your continued interest in our research.

Best regards,
Shengdong Nie

Round 3

Reviewer 2 Report

The authors exhaustively answered most of my questions from the previous review and introduced a lot of satisfactory corrections and additions.

However, four issues remain. One of them is editorial, and three are substantive.

1. Table should be corrected 2. Giving r and R2 in one table doesn't make any sense. It's redundant information. Given r, you can easily calculate R2, and with R2 - calculate r. Why, as I suggested, the authors did not provide the actual level of statistical significance of the examined relationship, i.e. p?

2. I am completely unconvinced by the argument that statistical indicators collected and analyzed by relevant offices were omitted from the analyzes due to probable errors and their poor comparability. The collection of standardized statistical data is the basis for the good functioning of any developed country. This is how it is in my country and I can't imagine it being any different in China. Without it, good management and planning is impossible.

3. I also doubt the assurances that water transport does not play a major role in the economic links between the cities of the Pearl River agglomeration, many of which are port cities located on the seashore or along the course of large rivers and the great estuary. This was certainly one of the most important factors in the location of these cities and their long-term development. Perhaps its importance is now slightly less than in the past, but it is certainly not marginal. Just look at the data presented at: https://www.statista.com/statistics/1039268/china-pearl-river-waterway-passenger-transport-volume-quarterly/. This aspect should at least be indicated in the text of the article.

4. I also consider that the proof that the "indicator" of economic distance Dij satisfies the fourth axiom of metric space (triangle inequality) is false. It doesn't look like a metric. It is probably a metric only in some specific cases and I believe that it should be explicitly stated in the text. On the other hand, however, I do not think that it is of fundamental importance for the obtained results..

- If the expression (1+e^-x) is replaced with zero, then the left side of the equation is 0 and the whole thing doesn't make sense.

- If, in turn, (e^-x) is replaced with zero, it means that the Ljk value is multiplied by 1.

- Root z (roiroj) is zero, if one of the elements is zero, if both parts are sqrt(rojrok) we also replace it with zero, it means that at least two densities are zero and then sqrt(rojrok) is also zero.

- A contradiction also arises when the left side of the equation is replaced by lower bounds and the right side by upper bounds. That is not true.

I have no comments.

Author Response

(The authors gave the same response as above.)

Round 4

Reviewer 2 Report

The authors of the peer-reviewed article kindly agreed with my comments on the previous version and made some changes to the text. Although I am not 100% satisfied with these changes, I believe that the article can be published.